# The Association between Malnutrition and Oral Health in Older People: A Systematic Review

**DOI:** 10.3390/nu13103584

**Published:** 2021-10-13

**Authors:** Yne Algra, Elizabeth Haverkort, Wilhelmina Kok, Faridi van Etten-Jamaludin, Liedeke van Schoot, Vanessa Hollaar, Elke Naumann, Marian de van der Schueren, Katarina Jerković-Ćosić

**Affiliations:** 1Research Group Innovations in Preventive Health Care, HU University of Applied Sciences, 3584 Utrecht, The Netherlands; yne.fennema@gmail.com (Y.A.); liesbeth.haverkort@hu.nl (E.H.); seline.kok@hu.nl (W.K.); liedeke.vanschoot@hu.nl (L.v.S.); 2Research Support, Medical Library AMC, Amsterdam UMC-Location AMC, University of Amsterdam, 1105 Amsterdam, The Netherlands; f.s.vanetten@amsterdamumc.nl; 3Research Group Nutrition, Dietetics and Lifestyle, HAN University of Applied Sciences, 6525 Nijmegen, The Netherlands; vanessa.hollaar@han.nl (V.H.); e.naumann@han.nl (E.N.); Marian.devanderSchueren@han.nl (M.d.v.d.S.); 4Division of Human Nutrition and Health, Wageningen University and Research, 6708 Wageningen, The Netherlands

**Keywords:** malnutrition, undernutrition, oral health, dental status, older people, elderly people, systematic review

## Abstract

The aim of this systematic review was to examine the association between malnutrition and oral health in older people (≥ 60 years of age). A comprehensive systematic literature search was performed in four databases (PubMed, CINAHL, Dentistry and Oral Sciences Source, and Embase) for literature from January 2000 to May 2020. Both observational and intervention studies were screened for eligibility. Two reviewers independently screened the search results to identify potential eligible studies, and assessed the methodological quality of the full-text studies. A total of 3240 potential studies were identified. After judgement for relevance, 10 studies (cross-sectional (*n* = 9), prospective cohort (*n* = 1)) met the inclusion criteria. Three studies described malnourished participants as having fewer teeth, or functional (tooth) units (FTUs), compared to well-nourished participants. Four studies reported soft tissue problems in malnourished participants, including red tongue with blisters, and dry or cracked lips. Subjective oral health was the topic in six studies, with poorer oral health and negative self-perception of oral health in malnourished elderly participants. There are associations between (at risk of) malnutrition and oral health in older people, categorized in hard and soft tissue conditions of the mouth, and subjective oral health. Future research should be focused on longitudinal cohort studies with proper determination of malnutrition and oral health assessments, in order to evaluate the actual association between malnutrition and oral health in older people.

## 1. Introduction

Aging is a complex phenomenon that, partially due to the occurrence of chronic diseases, can result in frailty, limited mobility, and other aspects of physical and cognitive decline [1,2,3]. Major concerns for older people are poor general health and poor nutrition [1,3]. In the Netherlands, it is estimated that one in three older people receiving formal home care is malnourished, and nearly 20% of the independently living older people (>85 y) suffer from a poor nutritional status [4]. Prevalence rates of high risk for malnutrition in older people in Europe are 28% (hospital), 17.5% (residential care), and 8.5% (community settings), according to a recent systematic review [5]. Malnutrition risk is associated with older age, presence of disease, and gender [5]. Consequences of malnutrition include reduced immunity, frequent infections, overall physical and psychological decline, and higher mortality [6,7,8].

Malnutrition can be defined as “a state resulting from lack of intake or uptake of nutrition that leads to altered body composition (decreased fat free mass) and body cell mass leading to diminished physical and mental function and impaired clinical outcome from disease” [9]. Malnutrition is often demonstrated by reductions in body weight and body mass index (BMI), primarily as a result of inadequate nutritional intake of proteins and/or energy from calories [7]. Malnutrition can also be present in persons with normal body weight or overweight, when there is a substantial reduction in fat-free mass (FFM), also called sarcopenia.

In addition to the risk of malnutrition, older people also have an increased risk of developing oral health problems [10]. Worldwide, the burden of oral diseases in older people is growing, as there are high levels of tooth loss, periodontal disease, xerostomia, dental caries, and oral cancer [11]. Several studies have shown an association between malnutrition/poor nutritional status and oral health in older people [10,12,13,14,15,16]. Poor oral health can cause oral pain, chewing problems, periodontal disease, and tooth loss, which have a negative impact on nutritional intake, leading to poor nutritional status and risk of malnutrition. Inadequate intake of micronutrients and macronutrients can, in turn, lead to an increased risk of oral health problems such as gum disease, caries, and hyposalivation [17,18,19].

The association between nutritional status and oral health in older people seems evident. There is tentative evidence indicating a negative association between malnutrition and oral health, according to several systematic reviews [20,21,22]. Still, the association seems general, without focusing on detailed information about observed oral health conditions in terms of hard tissues of the mouth (e.g., dental caries, decayed/missing/filled teeth), soft tissues of the mouth (e.g., periodontitis, gingivitis), hyposalivation or xerostomia, and (general) subjective oral health (e.g., oral hygiene, oral-health-related quality of life, autonomy for oral care) in malnourished older people. Prevention of malnutrition and optimizing oral health conditions in older people can result in better overall health, increased self-dependency, and higher quality of life.

The aim of this systematic review was to examine the association between malnutrition and oral health in terms of hard and soft tissue conditions of the mouth, xerostomia and salivary flow, and general (subjective) oral health in older people (≥ 60 years of age).

## 2. Materials and Methods

This systematic review was conducted in adherence to the guidelines of the Preferred Reporting Items for Systematic Reviews and Meta-Analyses (PRISMA) statement.

### 2.1. Search Strategy

Four electronic databases—PubMed, Embase (Ovid), CINAHL (EBSCO), and Dentistry and Oral Sciences Source (DOSS)—were searched for literature from January 2000 to May 2020 with a combination of MeSH terms, terms in titles or abstracts (TIAB), free-text terms, and synonyms. Since approximately 2000, there has been an increase in published literature about malnutrition and oral health in PubMed. Therefore, studies were included from January 2000 to May 2020. In view of the exploratory character of this systematic review, the following MeSH terms were used: oral health, mouth diseases, jaw diseases, tooth diseases, taste disorders, dentition, malnutrition, nutritional status, sarcopenia, aged, and geriatrics. The search strategy of this systematic review is available from the first author (Y.A.) upon request.

Two reviewers (Y.A. and W.K.) independently screened the studies for eligibility. Only when a consensus was not possible was a third reviewer (E.H.) consulted. Reference lists of every selected publication were also screened for eligibility using the same procedure.

### 2.2. Selection Criteria for Studies

In view of the exploratory character of this systematic review, observational and interventional studies were included. Case reports, expert opinions, conference meetings, animal studies, summaries, papers, overviews, and reviews were excluded. The search was limited to the English and Dutch languages. Articles were eligible for inclusion only if they (1) described malnutrition, (2) described oral health, and (3) described the association between both. The definition and determination of both malnutrition and oral health were defined beforehand in order to assess the eligibility of the studies.

There are several screening and assessment tools for malnutrition. For this systematic review, malnutrition—or risk of malnutrition—had to be determined based on at least one or more anthropometric measures (BMI, weight loss, or fat-free mass), preferably together with the use of a validated nutritional screening or assessment tool for older adults (e.g., Short Nutritional Assessment Questionnaire (SNAQ), Malnutrition Universal Screening Tool (MUST), Subjective Global Assessment (SGA), Mini Nutritional Assessment (MNA), or MNA Short Form (MNA-SF)).

Oral health was defined as the condition of hard and soft tissues of the mouth, hyposalivation, xerostomia, and general (subjective) oral health (oral hygiene, mouth pain, oral-health-related quality of life (OHRQoL)). Hard tissues of the mouth comprises he mineralized tissues: alveolar bone (jaw bone), enamel, root cement, and natural teeth [23]. Soft tissues of the mouth comprise the mouth membranes: gingiva, alveolar mucosa, periodontal ligament, and mucous membranes [23], with common oral diseases including periodontitis, gingivitis, candidiasis, and denture stomatitis [24,25]. Xerostomia is the (subjective) feeling of dry mouth, while hyposalivation refers to an objectively measured lower salivary rate [26].

### 2.3. Selection Criteria Populaton

The population of interest was older people, of 60 years or older. Studies were excluded if participants (1) had cancer or malignancies, (2) were terminally ill, (3) had dysphagia or chewing problems due to medical conditions such as cerebral vascular accident or musculoskeletal disease, or (4) received (complete) enteral or parenteral tube feeding.

### 2.4. Methodological Quality Assessment

Methodological quality was assessed independently by two reviewers (Y.A. and W.K.) using the Newcastle–Ottawa Scale (NOS) for non-randomized trials and observational studies [27]. Assessment was based on selection, comparability, and outcome, with a maximum score of 10. Studies with a score of 6 or less were excluded to guarantee the quality of the included studies. A level of evidence was adjudged to each article by two reviewers independently (Y.A. and W.K.). Any disagreement between the two reviewers about assigning a methodological quality score and level of evidence to the articles was resolved through discussion with a third review author (E.H.).

### 2.5. Data Extraction

Information with regard to study design, population, measures, and outcomes was extracted from the included studies by one reviewer (Y.A.), and reviewed by a second reviewer (W.K.). This information included first author, year, country, number and mean age of participants, setting, measures of malnutrition, and oral health status.

### 2.6. Clinical and Methodological Heterogeneity

Study results were summarized using descriptive statistics. Meta-analysis was impossible due to clinical and methodological diversity as a result of the broadly formulated research question, the various definitions of malnutrition and oral health in the included studies, and the variability of the participants, measurements, outcomes, and study designs.

## 3. Results

### 3.1. Study Selection

The electronic database search resulted in 3240 studies, refined to 1988 potential studies after removing duplicates. Titles and abstracts were screened based on study design, population, reporting on malnutrition and oral health, and outcome measures (malnutrition or oral health). After screening of titles and abstracts, 207 studies remained to be reviewed in their full text, out of which 195 did not meet the selection criteria and were excluded, resulting in a total of 12 potentially eligible studies (Figure 1). An overview of studies (references) excluded, by reason for exclusion, is available upon request from the first author (Y.A.).

### 3.2. Methodological Quality of the Studies

Eleven studies had a cross-sectional design and one study was a longitudinal cohort study. The methodological quality scores from these studies ranged from 5 to 10. Two studies were excluded due to having an NOS methodological quality score less than 7. Ten studies were included in this systematic review, with a mean quality score of 8.9 ± 0.54 (Appendix A).

### 3.3. Study Characteristics

General study characteristics are presented in Table 1. The 10 included studies had a cross-sectional design (*n* = 9) or longitudinal design (*n* = 1), with wide variation in outcome variables and statistics (mean and standard deviations, percentages, odds ratio or hazard ratio with 95% confidence interval, *p*-values). The numbers of participants in the studies ranged from 159 to 3320, with a total number of 9093 participants in this systematic review. In most of the studies, the proportion of women was higher than that of men. Participants were derived from different settings: nursing homes, dental clinics, community -welling older people, or hospital units (acute care units or rehabilitation). The prevalence of malnourished participants in the studies varied from 11.7% to 60%, and the reported range of participants “at risk of malnutrition” was 21–60%.

Malnutrition was assessed via the Mini Nutritional Assessment (MNA) (*n* = 4), MNA Short Form (MNA-SF) (*n* = 1), Subjective Global Assessment (SGA) (*n* = 1), body mass index (BMI) in combination with unintentional or time-specific weight loss (*n* = 2), or MNA in combination with BMI and weight loss (*n* = 1). Sarcopenia was assessed by low handgrip strength and/or low gait speed, with CC measurement of <33 (female)/<34 (male) (*n* = 1).

Several measurement instruments were used to assess subjective and objective oral health aspects, or a combination of both. Objective oral health was evaluated by using the Revised Oral Assessment Guide (ROAG) or ROAG-Jönköping (ROAG-J) (*n* = 2), oral examination or evaluation by a professional (*n* = 3), or the Oral Health Assessment Tool (OHAT) (*n* = 1). Additional data on objective oral health were reported with regard to the use of dentures (*n* = 4), decayed/missing/filled teeth (DMFT) or decayed/filled teeth (DFT) (*n* = 3), functional tooth units (FTUs) or functional units (FUs) (*n* = 3), number of teeth (*n* = 2), stimulated salivary flow (*n* = 2), and gingival inflammation (*n* = 1). Subjective oral health was assessed using the Geriatric Oral Health Assessment Instrument (GOHAI) (*n* = 2), self-administered or standardized questionnaires (*n* = 2), and the Oral Health Impact Profile (OHIP) (*n* = 1). Additional data on subjective oral health were reported regarding xerostomia (*n* = 2), chewing problems *(n* = 3), general mouth problems (*n* = 2), pain (*n* = 1), oral candidiasis (*n* = 1), and oral hygiene (*n* = 2).

### 3.4. Malnutrition and Hard Tissue Conditions of the Mouth

Three studies explored the number of dental functional units (Table 2). The number of functional tooth units (FTUs) was defined as pairs of upper and lower opposing natural teeth and/or artificial teeth on removable or fixed dentures [35,37]. Functional units (FUs) were defined as a pair of posterior antagonist teeth with at least one contact area during chewing [29]. Two studies showed that malnourished older people had significantly less FUs (<4) [29] or FTUs (8.3 ± 1.1) [35] compared to older people without malnutrition. In one study, there were no significant differences in terms of FTUs between older people with and without sarcopenia [37].

Two studies explored the associations based on number of teeth (Table 2). Samnieng et al. [35] reported a significantly increased malnutrition risk in older people with fewer teeth, compared to older people with more teeth. A second study demonstrated significantly lower numbers of teeth in older people with sarcopenia, compared to those without sarcopenia [37]. On the other hand, Mesas et al. [33] reported a non-significant association between the oral health indicator “edentulous” and (at risk of) malnutrition. In addition, Andersson et al. [28] found that the mean values for decayed/missing/filled teeth (DMFT) and decayed/filled teeth (DFT) did not differ between malnourished older people and older people with normal nutritional status. Furthermore, the associations between nutritional status and both DMFT and prosthetic status were not significant (*p* > 0.05).

### 3.5. Malnutrition and Soft Tissue Conditions of the Mouth

Conditions with regard to soft tissues of the mouth related to malnutrition were presented in four studies (Table 3). Malnourished participants had a higher proportion of oral problems in soft tissues. This included, for example, a tongue with no papillae, white coating, blisters, or ulceration. According to the ROAG items, lips were dry or cracked, and gums were edematous or red. Moreover, dry, red tick mucous membranes or ulcerations with bleeding were spotted [28,32]. Poisson et al. [34] reported low salivary flow in malnourished older people. Malnourished older people had the highest proportion of oral problems in soft tissues according to Lindmark et al. [32].

Andersson et al. [28] presented a significant association between malnutrition and tongue problems according to the ROAG. The study of Mesas et al. [33] found a significant association between the nutritional deficit (MNA score < 24 points) and advanced periodontal disease (defined as at least one sextant with pocket depths ≥ 6 mm). The presence of candidiasis in the mouth was associated with malnutrition as estimated by low MNA scores, according to Poisson et al. [34].

### 3.6. Malnutrition and Hyposalivation or Xerostomia

Conditions with regard to hyposalivation or xerostomia were presented in eight studies (Table 4).

Hyposalivation: Malnourished older people had the highest proportion of oral problems, including low salivary flow [32,34]. In the study of Andersson et al. [28], low salivary flow according to the ROAG was associated with the presence of malnutrition. Stimulated salivary flow rate (hyposalivation) was measured in the study of Mesas et al. [33], and a salivary flow rate < 0.7 (mL/min) was associated with nutritional deficit (MNA score < 24 points).

Xerostomia: Weak and non-significant associations between malnutrition and xerostomia were demonstrated by Huppertz et al. [30]. However, three studies showed significant associations between malnutrition and xerostomia. According to El Osta et al. [29], participants with xerostomia were more likely to be malnourished. Kiesswetter et al. [31] demonstrated associations between xerostomia and incident malnutrition. In addition, xerostomia was more pronounced in people without teeth and with incident malnutrition. In the study of Soini et al. [36], malnutrition increased consistently with the increasing number of oral health problems (including chewing problems, oral pain, and xerostomia).

### 3.7. Malnutrition and Subjective Oral Health

Eight studies demonstrated associations between malnutrition and other (general) subjective oral health indicators (Table 5). In the study of El Osta et al. [29], discomfort when eating, trouble biting/chewing, and lower mean Additive Geriatric Oral Health Assessment Instrument (ADD-GOHAI) scores were associated with malnutrition (*p* < 0.0001). Of the malnourished participants in the study of Huppertz et al. [30], 58.8% complained of poor oral health and 24.3% complained of general mouth problems (not further specified). According to Soini et al. [36], lower MNA values had a significant relationship with the number of oral health problems, such as chewing problems, swallowing difficulties, and oral pain. Lindmark et al. [32] demonstrated at least one oral health problem in one-third of the older people at risk of malnutrition; problems were seen in lips, mucous membranes, tongues, and saliva (*p* < 0.001). Mesas et al. [33] reported an association between nutritional deficit (MNA score < 24 points) and negative self-perception of oral health. Poisson et al. [34] reported that decreases in autonomy of oral care were independently associated with malnutrition according to the MNA (*p =* 0.004). The study of Takahashi et al. [37] demonstrated sarcopenia to be an independent exploratory factor of OHIP-14 scores. However, Kiesswetter et al. [31] reported that no differences were found between groups with and without malnutrition, with regard to their self-perceived oral health characteristics (relating to teeth, dentures and oral hygiene).

## 4. Discussion

This systematic review describes the associations between malnutrition and hard and soft tissue conditions of the mouth, hyposalivation, xerostomia, and subjective oral health in older people. Ten cross-sectional studies were included, and demonstrated an association between malnutrition and poor oral health conditions. Five studies indicated an association between number of F(T)Us and number of teeth, while one study described this association for edentulism and malnutrition. Four studies demonstrated associations between different soft tissue conditions of the mouth in malnourished older people. Periodontal disease, candidiasis, red or bleeding gums, blisters, tongue problems, and dry or cracked lips were more frequently present in malnourished older people, compared to older people with normal nutritional status. The association between (stimulated) low salivary flow or xerostomia and malnutrition was demonstrated in four studies. Seven studies reported subjective oral health aspects that were associated with malnutrition, or were more frequently reported by malnourished participants, such as pain (when chewing), autonomy of oral care, and negative self-perception of oral health.

These results confirm the previously demonstrated association between malnutrition, or nutritional status, and oral health problems in older people [11,13,14,15,16,17].

### 4.1. Measurement and Definition of Malnutrition and Oral Health

In 2018, consensus criteria were published by the Global Leadership Initiative on Malnutrition (GLIM) to define malnutrition in adults. However, there still is variation in the definition of malnutrition. The studies included in this systematic review frequently used nutritional assessment tools, such as the MNA.

Although BMI is often not the most valid method to determine malnutrition, the published literature demonstrates significant associations with oral health problems. Studies reported an association between low BMI and tooth loss, BMI and dry mouth when eating, and association between underweight and dryness, pain, uncomfortable sores, and irritation in the mouth [38,39,40,41].

Similarly to malnutrition, oral health is not unambiguously measured. Several measurement instruments were used to assess oral health, based on questionnaires and/or oral clinical assessment performed by a dentist or a nurse. The review of Everaars et al. [42] showed the methodological limitations of the available oral health assessments for non-dental healthcare professionals, and the limited quality of their measurement properties. Furthermore, the difference between hyposalivation and xerostomia should be stressed here. Xerostomia is the (subjective) feeling of dry mouth, and it can also occur while someone has normal salivation [26], while hyposalivation refers to an objectively measured lower salivary rate. As these two terms are sometimes used interchangeably, in some of the included studies it is not clear how the information about xerostomia was collected.

### 4.2. Methodological Limitations of Included Studies

Firm conclusions based on this systematic review are hampered by the statistical and clinical heterogeneity of the included studies, as well as the cross-sectional design of the included studies. All studies had a cross-sectional design, with low levels of evidence, measurement at one timepoint, and no control group or intervention. The methodological quality of the studies varied from 8 to 10 (mean score of 8.9 ± 0.54), with the lowest scores for complete description regarding methodological processes, non-respondents, comparability of the subjects, controlling for confounding, and statistical tests. Adjustment for confounding was described in a few studies; however, with the use of (partially) self-reported data in cross-sectional studies (subjective oral health, xerostomia, weight loss, or BMI), confounding factors and measurement bias can be a problem.

### 4.3. Strengths and Limitations

This systematic review was strengthened by the broadly formulated inclusion criteria regarding study designs and types of participants. Moreover, a large number of participants from different settings were included, with ages of 60 years or older. However, only a few studies included community-dwelling older people, which suggests that the studies are not representative of the general older population.

A comprehensive search strategy was conducted in four electronic databases, with search strategy support from two medical librarians. In all of the included studies, malnutrition was the actual outcome, and not a surrogate outcome. Furthermore, the literature was independently evaluated by two reviewers, as was the methodological quality of the potential eligible studies. Older (published before the year 2000) systematic reviews on the topic of malnutrition and oral health were examined to ensure that no relevant literature was excluded beforehand. Finally, we acknowledge the risk of publication bias. The 10 included studies often presented significant associations. However, publication bias is difficult to identify.

### 4.4. Implications for Practice

In this systematic review, malnourished older people had significantly more impaired soft tissue conditions related to mucous membranes, periodontium, gums, and tongue, such as a tongue with no papillae, white coating, blisters, or ulceration. Oral health appears to be important in nutritional care. Preventive healthcare and multidisciplinary cooperation will be important with regard to the aging population over the next few decades. The knowledge from this systematic review can contribute to the development of screening instruments and guidelines. In addition, it can help healthcare professionals to better identify problems in the field of malnutrition and oral health. Reducing the prevalence or severity of malnutrition and oral health conditions in older people can result in better overall health, increased self-dependency, and higher quality of life.

## 5. Conclusions

Despite the limitations of this review, there are indications that the presence of malnutrition is related to the state of hard and soft tissues of the mouth, salivary flow, and xerostomia, as well as other subjective oral health aspects. In the end, there is an extensive interrelation between oral health and malnutrition; however, it remains unclear whether this is a two-way association, or whether poor oral health increases the risk of being malnourished, or vice versa: that being malnourished results in poor oral health in older people.

Future research should be focused on longitudinal cohort studies with proper determination of malnutrition and oral health assessments, so as to evaluate the actual association between malnutrition and oral health in older people.

## Figures and Tables

**Figure 1 nutrients-13-03584-f001:**
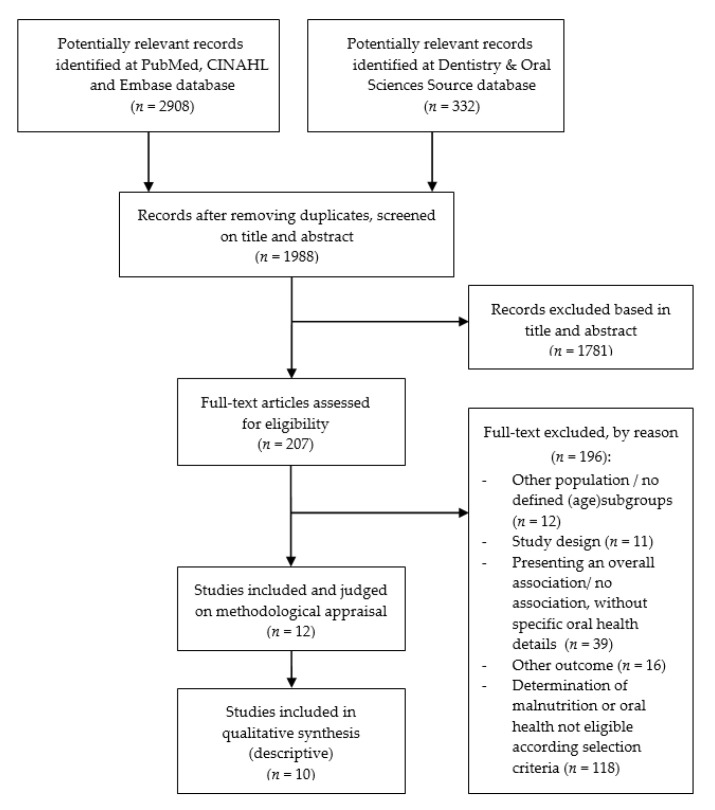
Flow chart of study selection.

**Table 1 nutrients-13-03584-t001:** Main study characteristics.

First Author	Publication Year, Country	Number of Participants	Mean Age	Setting	Measurement of Malnutrition	Measurement of Oral Health	Methodological Quality Score
Andersson [28]	2004, Sweden	*n* = 161 M: 43 F: 118	81.7 (range 65–89)	Three rehabilitation wards at a university hospital	SGA	ROAG	9
El Osta [29]	2013, Lebanon	*n* = 201 F: 121 M: 80	F: 71.6 ± 6 M: 72.7 ± 7	Older people attending two acute care units	MNA	GOHAI, DMTF/DTF, prosthetic status, posterior dental FUs	9
Huppertz [30]	2017, The Netherlands	*n* = 3320 M: 1059 F: 2261	84.3 ± 7.4	Nursing homes	BMI in combination with unintentional weight loss	Standardized questionnaire on potential indicators of poor oral health	9
Kiesswetter [31]	2019, The Netherlands	*n* = 893 M: 418 F: 475	67.6 ± 6.1	Community-dwelling older people	BMI, time-specific weight loss	Self-administered questionnaire (22 items) on four oral health domains	8
Lindmark [32]	2017, Sweden	*n* = 1156 M: 443 F: 713	82.8 ± 7.9	Nursing homes and hospitals	MNA-SF	ROAG-J	8
Mesas [33]	2010, Sweden	*n* = 267 M: 107 F: 160	66.5 ± 4.1	Community-dwelling older people	MNA	GOHAI, number of teeth, prosthesis, posterior occlusion, stimulated salivary flow, CPITN	10
Poisson [34]	2014, France	*n* = 159 M: 51 F: 108	85.3 ± 5.7	Patients hospitalized in acute care units: 78% (*n* = 124) from home, 22% (*n* = 35) from nursing homes	Weight loss, BMI, MNA	Oral examination by dentist, DMTF index, gingival inflammation, oral candidiasis, salivary test (insufficiency if salivary flow < 0.1 g/min/weight compress < 0.35 g)	9
Samnieng [35]	2011, Thailand	*n* = 612 M: 158 F: 454	68.8 ± 5.9	Community-dwelling older people	MNA	Dental status assessed by dentist; DMFT, prostheses, FTUs	9
Soini [36]	2006, Finland	*n* = 3088 M: 649 F: 2439	NH: 81 LT: 83	Institutionalized older people from NH (*n* = 2036) and LT (*n* = 1052)	MNA	Oral status evaluated by trained ward nurses	9
Takahashi [37]	2018, Japan	*n* = 279 M: 106 F: 173	76 ± 7.5	Older people at a dental clinic	Sarcopenia (GS, HS, MNA-SF, BMI, EAT-10, CC).	Number of teeth, FTUs. Primary outcome: OHIP to evaluate OHRQoL. Secondary outcome: OHAT	9

Level of evidence of all studies level 4. Legend: ±: standard deviation; ADD-GOHAI: Additive Geriatric Oral Health Assessment Instrument; BMI: body mass index; CC: calf circumference; CPITN: periodontal condition; DMTF: decayed/missing/filled teeth; DFT: decayed/filled teeth; EAT-10: 10-item Eating Assessment Tool; F: female; FUs: (posterior dental) functional units; FTUs: functional tooth units; g: gram; GOHAI: Geriatric Oral Health Assessment Instrument; GS: gait speed; HS: handgrip strength; LT: long-term care wards; M: male; MNA: Mini Nutritional Assessment; MNS-SF: Mini Nutritional Assessment Short Form; N: number; NH: nursing homes; OHAT: Oral Health Assessment Tool; OHRQoL: oral-health-related quality of life (OHRQoL); OHIP: Oral Health Impact Profile; ROAG: Revised Oral Assessment Guide; ROAG-J: Revised Oral Assessment Guide-Jönköping; SGA: Subjective Global Assessment.

**Table 2 nutrients-13-03584-t002:** Outcome measures regarding hard tissue conditions in malnourished older people compared to well-nourished older people.

Item	[Ref]	Prevalence (N(%)), or No. of Teeth/FU		
		MN	At Risk of MN	Well-Nourished
No. of FUs: <4 FUs 5 or 6 FUs 7 or 8 FUs	[29]	*n* total = 85 *n* = 52 (61%) ** *n* = 6 (7.1%) *n* = 27 (31.8%)	- - -	*n* total = 116 *n* = 38 (32.8%) ** *n* = 24 (20.7%) 54 (46.6%)
No. of decayed teeth	[35]	1.6 ± 0.3 *	1.3 ± 0.1 *	1.1 ± 0.2 *
No. of teeth	[35] [37]	8.7 ± 1.4 * 13.4 ± 9.3	10.1 ± 0.4 * -	13.2 ± 0.7 * 18.9 ± 7.8
No. of FTUs	[35] [37]	8.3 ± 1.1 * 10.0 ± 3.5	8.4 ± 0.3 *	10.3 ± 0.5 * 10.5 ± 2.5
Association ND and edentulism	[33]	Crude OR: 1.44 ^a^ (95% CI 0.61–3.33) Adjusted OR ^b^: 0.65 (95% CI 0.23–1.83)		

Legend: *: *p* < 0.05; **: *p* < 0.0001; ±: standard deviation; ^a^: chi-squared test or Fisher’s exact test; ^b^: logistic regression of the association between each indicator of oral health and nutrition deficit adjusted for gender, age, schooling, economic class, smoking, depression, and medication use; CI: confidence interval; FUs: (posterior dental) functional units; FTUs: functional tooth units; MN: malnourished; ND: nutritional deficit; No.: number.

**Table 3 nutrients-13-03584-t003:** Outcome measures regarding soft tissue conditions in malnourished older people compared to well-nourished older people.

Item	[Ref]	Prevalence’s		
		MN	At Risk of MN	Well-Nourished
Tongue	[32] [28]	40 (20.3%) 43 (49%)	38 (7.4%) -	21 (4.7%) -
Mucous membranes	[32] [28]	41 (21.3%) 26 (30%)	37 (7.2%) -	14 (3.2%) -
Lips	[32] [28]	35 (17.8%) 48 (55%)	26 (5.0%) -	17 (3.8%) -
Gums	[32] [28]	26 (14.4%) 14 (16%)	42 (8.7%) -	20 (5.0%) -
Candidiasis	[34]	12 (15.6%) *p* < 0.001	-	-
Association MN and tongue problems	[28]	OR 4.4 (95 % CI 2.0–9.6; *p* < 0.0005)		

Legend: CI: confidence interval; MN: malnourished; OR: odds ratio; *p*: *p*-value

**Table 4 nutrients-13-03584-t004:** Outcome measures regarding hyposalivation and xerostomia in malnourished older people compared to well-nourished older people.

Outcomes	First Author [Ref]
Hyposalivation	
Association nutritional deficit and stimulated salivary flow < 0.7 mL/min: Crude OR 1.96 (95% CI 1.06–3.83) Adjusted OR 2.18 (95% CI 1.06–4.50).	Mesas [33]
Association salivary flow rate < 0.7 (mL/min) and nutritional deficit: Adjusted OR 2.18 (95% CI 1.06–4.50).	Poisson [34]
Xerostomia	
Perception of xerostomia as parameter of explaining MNA variation: OR 3.49 (95% CI 1.657–7.337; *p* = 0.001).	El Osta [29]
Association between xerostomia and incident malnutrition HR 2.63 (95% CI 1.18–6.26).	Kiesswetter [31]

Legend: CI: confidence interval; HR: hazard ratio; MNA: Mini Nutritional Assessment; OR: odds ratio; *p*: *p*-value.

**Table 5 nutrients-13-03584-t005:** Outcome measures regarding subjective oral health.

Outcomes and Prevalence: Subjective Oral Health	First Author [Ref]
Negative self-perception of oral health: Crude OR: 3.95 (95% CI 2.04–7.67) Adjusted OR: 3.41 (95% CI 1.59–7.33)	Mesas [33]
OHRQoL/oral status: Poorer OHRQoL and oral health status (all *p* < 0.001). GOHAI score explains MNA variation: OR: 2.905 (95% CI 1.40–6.00; *p* = 0.004).	Takahashi [37] El Osta [29]
OHRQoL/oral status: Negative correlation between the ROAG-J total score and MNA total score (r = −0.241; *p* < 0.001).	Lindmark [32]
Association between toothache while chewing (adjusted) HR 2.14 (95% CI 1.10–4.19; *p* = 0.026).	Huppertz [30]

Legend: CI: confidence interval; GOHAI: Geriatric Oral Health Assessment Instrument; HR: hazard ratio; MNA: Mini Nutritional Assessment; OHRQoL: oral-health-related quality of life; OR: odds ratio; *p*: *p*-value; ROAG-J: Revised Oral Assessment Guide-Jönköping.

## Data Availability

Not applicable—because this is a systematic review, all data are available in the primary studies.

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
