# Peer review of "The Association between Malnutrition and Oral Health in Older People: A Systematic Review"

_nutrients, 2021, doi:10.3390/nu13103584_

Round 1
Reviewer 1 Report
Oral and nutritional issues are areas that must be pursued in future intervention studies. I think it's good to have been organized in this review.
I would like you to organize the table a little more. For example, organize by the same item,
Reviewer 2 Report
Please refer to the attached file

Reviewer 3 Report
The article submitted for the review is quite interesting and deals with the current issue of the elderly. The authors prepared the analysis of the issue thoroughly. However, after analyzing the article, I propose a correction:
- In reference 49, no bibliomertical data is available for articles (2019, volume 12; number 3: pages 1131-1135)
- Please explain item 130 of the reference, I guess it should be: Espinosa-Val MC, Martín-Martínez A, Graupera M, Arias O, Elvira A, Cabré M, Palomera E, Bolívar-Prados M, Clavé P, Ortega O. Prevalence, Risk Factors, and Complications of Oropharyngeal Dysphagia in Older Patients with Dementia. Nutrients. 2020 Mar 24;12(3):863. doi: 10.3390/nu12030863. PMID: 32213845; PMCID: PMC7146553.
- I propose to remove from the Supplementary Materials: Table S1: Studies excluded by reason.
Only articles qualified for review should be left in the reference list. In this form, with 229 references, the article is somewhat illegible. In my opinion, focusing only on the analyzed articles will facilitate the presentation of the article.
Round 2
Reviewer 2 Report
Table 2 and 3: I suggest that the author also report the prevalence of oral problems in the well-nourished group. Readers may be interested to know the difference in the prevalence rate of oral problems between well-nourished and malnourished.
Table 3: The definition of MN should be added in the footnote.
